# In Vitro and In Silico Study of the α-Glucosidase and Lipase Inhibitory Activities of Chemical Constituents from *Piper cumanense* (Piperaceae) and Synthetic Analogs

**DOI:** 10.3390/plants11172188

**Published:** 2022-08-24

**Authors:** Juliet A. Prieto-Rodríguez, Kevin P. Lévuok-Mena, Juan C. Cardozo-Muñoz, Jorge E. Parra-Amin, Fabián Lopez-Vallejo, Luis E. Cuca-Suárez, Oscar J. Patiño-Ladino

**Affiliations:** 1Departamento de Química, Facultad de Ciencias, Pontificia Universidad Javeriana, Bogotá 110231, Colombia; 2Departamento de Química, Facultad de Ciencias, Universidad Nacional de Colombia, Sede Bogotá, Bogotá 111321, Colombia; 3Facultad de Ciencias, Universidad de Ciencias Aplicadas y Ambientales, Bogotá 111166, Colombia; 4Departamento de Física y Química, Facultad de Ciencias Exactas y Naturales, Universidad Nacional de Colombia-Sede Manizales, Kilómetro 9 vía al aeropuerto, La Nubia, Manizales 170003, Colombia

**Keywords:** *Piper cumanense*, obesity, type 2 diabetes, digestive enzymes, pancreatic lipase, α-glucosidase

## Abstract

Digestive enzymes are currently considered important therapeutic targets for the treatment of obesity and some associated metabolic diseases, such as type 2 diabetes. *Piper cumanense* is a species characterized by the presence of bioactive constituents, particularly prenylated benzoic acid derivatives. In this study, the inhibitory potential of chemical constituents from *P. cumanense* and some synthesized compounds was determined on digestive enzymes (pancreatic lipase (PL) and α-glucosidase (AG)). The methodology included isolating and identifying secondary metabolites from *P. cumanense*, synthesizing some analogs, and a molecular docking study. The chemical study allowed the isolation of four prenylated benzoic acid derivatives (**1**–**4**). Four analogs (**5**–**8**) were synthesized. Seven compounds were found to significantly inhibit the catalytic activity of PL with IC_50_ values between 28.32 and 55.8 µM. On the other hand, only two compounds (**6** and **7**) were active as inhibitors of AG with IC_50_ values lower than 155 µM, standing out as the potential multitarget of these chromane compounds. Enzyme kinetics and molecular docking studies showed that the bioactive compounds mainly interact with amino acids other than those of the catalytic site in both PL and AG. This work constitutes the first report on the antidiabetic and antiobesity potential of substances derived from *P. cumanense*.

## 1. Introduction

The digestive enzymes related to the metabolism of lipids and carbohydrates such as pancreatic lipase and α-glucosidase are currently considered important therapeutic targets for the treatment of obesity and some associated metabolic diseases such as type 2 diabetes [1,2,3,4]. The use of pancreatic lipase inhibitors is a commonly used strategy to reduce fat absorption and lose weight [5,6], while the inhibition of α-glucosidase is important in postprandial glucose control because it can lead to the slower digestion of carbohydrates and a reduction in glucose absorption [1,7]. Some medicines are commonly used in clinical therapy to decrease hyperglycemia, hyperlipidemia, and obesity, among which acarbose and miglitol stand out as AG inhibitors and orlistat as a PL inhibitor. However, these products have been associated with different side effects (gastrointestinal problems including bloating, diarrhea, flatulence, and abdominal discomfort) and are considered to have low effectiveness for obesity and diabetes treatment [6,7,8]. Therefore, it is necessary to develop new, efficient, and safe agents to inhibit these types of enzymes that participate in the metabolism of carbohydrates and fats. In this sense, much research has focused on the natural resources to find more effective PL and AG inhibitors with minimal adverse effects because they produce a wide variety of chemical substances without medicinal chemistry exploration, many of which can inhibit digestive enzymes [9].

In the search for new therapeutic agents for drug discovery, species of the Piperaceae family represent an interesting and promising alternative due to the diversity of compounds they cover and their bioactive potential [10,11]. The genus *Piper* is the most representative of the family with around 1500 species distributed worldwide, mainly in tropical and subtropical regions [12]. *Piper* species have been used globally to treat several diseases including pathologies involved in the development and exacerbation of metabolic syndrome and its main components (central obesity, hyperglycemia, hypertension, and dyslipidemia) [13,14]. Among its most representative species used in traditional medicine for the treatment of obesity and diabetes are *P. nigrum*, *P. sarmentosum*, *P. longum*, and *P. beetle* [15,16,17]. Research carried out on these species has made it possible to validate traditional uses and identify the chemical constituents responsible for bioactivity. The *P. longum* root aqueous extract showed antihyperlipidemic and antihyperglycemic effects in STZ-induced rats after 30 days of treatment with a dose of 200 mg/kg body weight, causing a significant decrease in blood glucose level, total cholesterol, triglycerides, very low-density lipoprotein, and low-density lipoprotein, and an increase in the high-density lipoprotein [18]. The essential oil of fruits from *P. longum* has shown inhibitory activity on the enzymes AG, aldose reductase (AR), and PL, finding that the IC_50_ values were 150, 120, and 175 μg/mL, respectively [19]. The aqueous leaf extract of *P. sarmentosum* demonstrated antidiabetic and regulatory cardiovascular effects in Sprague Dawley rats after 28 days of treatment with 0.125 g/kg body weight of extract, causing a decrease in fasting blood glucose and urine glucose level and fewer ultrastructural degenerative changes in the cardiac tissues and the proximal aorta [20]. Another study revealed that the oral administration of 125 mg/kg body weight methanolic extract of *P. sarmentosum* to fructose-induced obese rats led to a significant amelioration of obesity and hyperlipidemia because it exerted protective effects by suppressing adipocytes that reduce obesity and inhibit HMG-CoA, which reduces hyperlipidemia [21]. The leaf extract of *P. sarmentosum* and its constituents of phenylpropanoyl amide type were able to inhibit AG, with chaplupyrrolidone B being the most active compound with an IC_50_ of 430 µM [22]. *P. sarmentosum* (PSAE) and *P. beetle* (PBAE) aqueous extracts at a 100 µg/mL concentration demonstrated PL inhibitory activity (PSAE = 67.4%; PBAE = 48.2%) and AG inhibitory activity (PSAE = 76.89%; PBAE = 51.65%) [23]. The *P. beetle* ethanolic leaf extract showed an inhibitory effect on AG (IC_50_ = 98.6 µg/mL) and α-amylase (AA) (IC_50_ = 26.2 µg/mL) [24]. The antidiabetic properties of *P. nigrum* have been confirmed *in vivo*; treatment with 100, 200, and 300 mg/kg body weight of the leaf methanolic extract of *P. nigrum* reduced blood glucose levels in alloxan-induced diabetic rats after 21 days of treatment [17,25]. Recently, the AA and AG inhibitory effect of ethanolic extract of fruits from *P. nigrum* and piperine was established, finding that piperine (IC_50_ = 216 µg/mL on AA and IC_50_ = 147 µg/mL on AG) showed a higher AA and AG inhibitory effect rather than the extract (IC_50_ = 105 µg/mL on AA and IC_50_ = 300 µg/mL on AG) [26].

*Piper cumanense* Kunth (synonyms *Artanthe cumanensis* (*Kunth*) *Miq.*, *P. sternii Yunck.*, *P. tolimae C. DC*., *P. variegatum Pers.*, *Schilleria cumanensis* (*Kunth*)) is a species that has been found in Latin America, mainly in Costa Rica, Panama, Colombia, Ecuador, Peru, Brazil, and Venezuela [27,28,29]. Previous research on the species reports antiparasitic potential [16,30,31,32], anticancer properties [33,34], and the potential to inhibit the growth of various phytopathogenic agents [35]. Bioactive chemical constituents have been identified in *P. cumanense*, including mainly prenylated benzoic acid derivatives, terpenes, and flavonoids [27,28,29,32]. The inhibitory effect on digestive enzymes of *P. cumanense* and its constituents has not been reported. This research aimed to characterize the inhibitory potential against PL and AG of chemical constituents from *P. cumanense* and synthetic analogs.

## 2. Results and Discussion

### 2.1. Phytochemistry

The phytochemical study carried out on the ethanolic extract from inflorescences of *P. cumanense* led to the isolation of four prenylated benzoic acid derivatives: (*2′E*) cumenic acid **1**, (*2′Z*) cumenic acid **2**, gaudichaudianic acid **3,** and 4-methoxy-3-(3′-methyl-2’-butenyl)benzoic acid **4** (Figure 1). The NMR spectra of the isolated compounds are presented in the Appendix A. These compounds have previously been identified in different organs from *P. cumanense* [29]. Compounds **1** and **2** have also been isolated from *P. gaudichaudianum* [35]; previous studies evaluated their antifungal activity against various phytopathogenic fungi [27]. Gaudichaudianic acid **3** has been reported in *P. gaudichaudianum* and *P. chimonantifolium* [36]. The antifungal activity of **3** has been evaluated against various phytopathogens, highlighting its action against *Cladosporium cladosporioides* and *C. sphaerospermum* [37]. Compound **4** was first reported in *Wigandia urens* (Boraginaceae) and its ability to bind to the chemokine receptor CCR5 has been confirmed [38]. The inhibitory action against digestive enzymes has not yet been reported for the compounds isolated from *P. cumanense*. However, the potential of oxyprenylated compounds against α-amylase, α-glucosidase, and lipase has recently been reported [39]. 

### 2.2. Synthesis of Analogs and Derivatives

The identified prenylated compounds from *P. cumanense* were used as a reference for the synthesis of compounds **5**–**8** (Figure 1) and thus established some preliminary structure–activity relationships. Compounds **5** and **6** were synthesized from a typical alkylation reaction on 4-chromanone in the presence of butyllithium [40]. The formation of the product of interest corresponding to 4-chromanone alkylated in the α position to the carbonyl group was not observed. Obtaining **5** may be supported by a retro-Michael reaction in which butyllithium promotes the opening of the pyran ring [41]. The formation of **6** may be favored by the nucleophilic addition of the butyl anion to the carbonyl group, generating its reduction to alcohol [42]. Compound **6** is reported for the first time, while **5** has been previously synthesized by a photo-Fries rearrangement reaction [43]. The route of synthesis described to obtain **5** and **6** has not been previously reported and represents an interesting alternative for the application of butyllithium in obtaining different alkylated products. Compound **7** was first synthesized from an electrophilic substitution reaction on a commercial chromene (6-bromo-2,2-dimethylchromene) in the presence of sec-butyllithium and followed by the addition of the prenylated aldehyde (citral) [41]. Compound **8** was first obtained in good yield by a typical methylation reaction on **1** in the presence of methyl iodide [44]. The NMR spectra of the synthesized compounds are presented in the Appendix A.

### 2.3. Inhibition of Digestive Enzymes

Four synthetic and four natural compounds isolated from the inflorescences of *P. cumanense* were tested using two in vitro enzyme assays to assess their PL and AG inhibition ability. Orlistat was used as a positive control in the PL inhibition assay, whereas acarbose was used for AG inhibition assays. The half-maximal inhibitory concentration (IC_50_), the inhibition constants (Ki) and the type of inhibition on each enzyme were estimated for the compounds that presented potential inhibitory activity (Table 1).

Regarding PL, the chromane-type compounds were characterized as the strongest inhibitors with IC_50_ values below 37 μM, with synthetic compound **6** being the most active. However, open-chain benzoic acid derivatives and aromatic ketones also showed promising activity on PL with IC_50_ values between 33.78 and 55.78 µM. Nevertheless, none of the compounds achieved comparable inhibition to that elicited by the positive control orlistat. Compound **4** did not show any activity. A preliminary structure–activity analysis indicates that the presence of oxoprenylated chains at position 3 of the benzoic acid nucleus is essential to inhibit PL since it is found that compounds **1**, **2** and **8**, which have the oxoprenylated chain, can inhibit the enzyme, while compound **4** does not have this chain and did not show activity on PL. It should be noted that the methyl ester of benzoic acid derivatives negatively influences the inhibitory effect on PL, a fact that is observed when comparing the IC_50_ values of compounds **1** and **8**. The kinetic study confirms that compounds **1** and **2** are uncompetitive, **5** and **6** are non-competitive, **8** is mixed, while **3** and **7** are competitive PL inhibitors (Appendix A). The affinity of the active compounds for the catalytic sites of the enzyme was estimated by experimental methods and corroborated with data obtained by web-based tools and through the Cheng–Prusoff equation. In this case, it was found that the estimated inhibition constants for compounds **1**, **2**, **3**, **6**, **7** and **8** are less than Km (47.40 µM); therefore, it is possible to affirm that the inhibitors have greater affinity with the enzyme than the substrate used in the assay. Compound **2** was the inhibitor with the highest affinity to PL (Ki = 7.88 μM), followed by its isomer **1** (Ki = 8.46 μM). This study constitutes the first report of inhibitory activity against PL for all the evaluated compounds.

On the other hand, regarding glycoside hydrolase enzyme, α-glucosidase, only two compounds were inhibitors, both chromane-type. As a result of the inhibition assays for this enzyme, chromane analogs **6** and **7** were shown to be more potent inhibitors than the positive control acarbose. According to the experimental conditions, the affinity of acarbose, a competitive inhibitor of the AG, is lower than that of chromane **6** (Ki = 51.30 μM), which has a mixed-type inhibition mechanism on this enzyme (Appendix A). Additionally, chromene **7** shows a lower affinity as an inhibitor for AG, with a Ki value of 135.81 μM. This study constitutes the first report of inhibitory activity against AG for all evaluated compounds.

### 2.4. Molecular Docking

Molecular docking studies of PL, AG, and active compounds (**1**–**3**,**5**–**8**) were carried out with AutoDock Vina and binding analysis visualization with Free Maestro and ICSF Chimera [45].

The binding site selection for each enzyme (orthosteric or allosteric) was guided by the experimental type of inhibition (Table 1) and blind molecular docking studies. Blind dockings were performed over the entire van der Waals surface of PL and AG proteins, and their results showed that, in both proteins, there are two binding sites where ligands bind with high affinities, as shown in Figure 2. The binding energies of active compounds are summarized in Appendix A.

Since the chiral compounds were biologically evaluated as racemic mixtures, it is important to assess whether their enantiomers differ in their binding modes and affinities. According to the kinetic studies, compounds **3** and **7** (both 2H-chromene) are PL competitive inhibitors and their docking results display high affinities for the active site (Appendix A). Molecular docking of ***3R*** and ***3S*** enantiomers show that they bind into the catalytic site of PL with almost the same binding mode; with a root means square deviations (RMSD) scaffold of 0.49 Å (Appendix A) and binding energies of −8.7 and −8.6 kcal/mol, respectively (Appendix A). Both enantiomers display short contacts with the catalytic site residues (His 263, Ser 152, Phe 77), with the main short contact being a π–π interaction between the chromane ring and the imidazole ring of catalytic residue His263 (Figure 3). As with compound **3**, ***7R*** and ***7S*** enantiomers show comparable binding modes with an RMSD scaffold of 0.36 Å (Appendix A) with binding energies of −8.9 and −8.7 kcal/mol, respectively (Appendix A). The slight difference between the binding energies of *R* and *S*, higher in *R* compared to *S*, could be explained by the hydrogen bond between the OH group attached to chiral carbon and the catalytic residue Ser152 (Figure 3). Interestingly, the alignment of the binding modes of compounds **3** and **7** shows that the chromene scaffold maintains a similar position (Appendix A) in the catalytic site of PL, thus confirming that the chromene scaffold is required for competitive inhibition of PL.

Kinetic studies reveal that compounds **1**, **2**, **5**, **6**, and **8** bind to a site other than the active site of PL; and molecular docking studies suggest that these compounds could bind into a pocket formed by residues from lipase and the cofactor colipase, which could modulate the allosteric regulation of PL (Figure 4). The benzoic acid derivatives **1**, **2**, and **8** share the binding mode (Appendix A) at the same binding pocket (Figure 4). These compounds form a hydrogen bond acceptor between the sp^3^ oxygen atom of the carboxyl and the NH group of the protein chain backbone at residue Asp387 of lipase, hydrogen bond donors with the side chains of Tyr369 and Arg337 of lipase, and additionally, make short contacts with Leu41, Lys42, and Glu64 of the cofactor colipase (Figure 4). Compounds **5** and **6** binds in a pocket located next to the pocket of compounds **1**, **2**, and **8**, where the predominant short contacts are made with cofactor colipase residues (Figure 4). It is worthy to mention that, although ***6R*** and ***6S*** enantiomers share the same binding pocket, they do not show the same binding mode (Appendix A). Allosteric inhibition of PL has also been reported elsewhere [46,47,48,49].

Regarding AG inhibitors, compounds **6** and **7** were the only active compounds and experimentally showed to bind to a site other than the active site. Blind docking studies of compounds **6** and **7** suggested at least three alternative binding sites to the catalytic site; this agrees with studies reported by Ding et al., where several allosteric AG sites were assessed by in silico and in vitro studies, and other authors have also reported allosteric regulation by synthetic and natural compounds [50,51,52]. However, in this work, only the binding site with the highest affinities was considered for further analysis (Figure 5). Compounds **6** and **7** are chromane and chromene, respectively. Although these compounds are structurally similar, they do not show similar binding modes, but the same binding site; the 8-carbon chain of compound **7** tends to occupy the pocket where the chromane ring of compound **6** binds (Appendix A). According to the docking results, the carbon chiral of compound **6** does not affect the binding modes of *R* and *S* enantiomers, making short contact mainly with polar residues such as Lys643, Lys647, Lys653, Lys765, and Glu757 (Figure 5 and Appendix A). Enantiomers ***7R*** and ***7S*** are slightly shifted from each other; however, they bind to the site with the same binding energy (−7.8 kcal/mol). Enantiomer ***7S*** displays two important interactions, a π–π and a hydrogen bond with the side chain of Arg653 (Figure 5 and Appendix A).

## 3. Materials and Methods

### 3.1. General Experimental Procedures

Thin-layer chromatography (TLC) was performed on SiliaPlate^TM^ alumina plates pre-coated with silica gel 60 F_254_ (SiliCycle^®^ Inc., Quebec, QC, Canada). Vacuum liquid chromatography (VLC) was performed on SiliaPlate^TM^ silica gel F_254_ of size 5–20 µm (SiliCycle^®^ Inc., Quebec, QC, Canada). Flash chromatography (FC) was performed on SiliaFlash^®^ silica gel P_60_ of size 40–63 µm (SiliCycle^®^ Inc., Quebec, QC, Canada). Melting points were recorded on a Thermo Scientific 00590Q Fisher-Johns apparatus (Thermo Scientific^®^, Waltham, MA, USA). IR spectra were determined on FT-IR Panagon 500 series 1000 spectrometer (PerkinElmer, Inc., Waltham, MA, USA). NMR measurements were performed on Bruker Advance AC-400 spectrometer (Bruker^®^, Hamburg, Germany). High-resolution mass spectrometry (HRMS) analyses were performed on LC-MS-TOF (Shimadzu Corporation, Kyoto, Japan) system. The ionization method was ESI operated in positive and negative ion mode. All commercially available reagents were used in the phytochemical and synthetic study without further purification, while technical grade solvents were distilled before use.

The enzymes used in this study were pancreatic lipase type II (PL) from porcine pancreas (100–400 units/mg protein, L-SLBD2433V, EC. 3.1.1.3, Sigma-Aldrich, San Luis, MO, USA) and α-glucosidase (AG) type I from Saccharomyces cerevisiae (lyophilized powder, ≥10 units/mg protein, L-SLBX6245, EC 3.2.1.20, Sigma-Aldrich, San Luis, MO, USA). 4-nitrophenyl dodecanoate (Sigma-Aldrich, San Luis, MO, USA) was used as the substrate for PL while 4-nitrophenyl-α-D-glucopyranoside (Sigma-Aldrich, San Luis, MO, USA) for AG. Absorbance readings were performed in a Multiskan GO microplate reader (Thermo Scientific^®^, Waltham, MA, USA).

### 3.2. Plant Material

The inflorescences of the specie were collected in Quipile town in the department of Cundinamarca (Colombia). *Piper cumanense* Kunth was determined by Adolfo Jara and a specimen was deposited in the Herbario Nacional Colombiano with voucher number COL-518185.

### 3.3. Extraction and Isolation

Air-dried and powdered inflorescences of *P. cumanense* (200 g) were extracted with EtOH 96% by the maceration method employing a ratio of 1:4 *w*/*v*, at room temperature for one week, changing the solvent every 48 h. The resulting solution was concentrated under a vacuum to obtain 23.5 g of extract. The ethanolic extract of inflorescences (23 g) was subjected to repetitive solid–liquid extraction with hexane, obtaining from the supernatant a crystalline yellow solid called (*2’E*) cumenic acid (**1**, 1.2 g, 112–113 °C). The residue of the extraction (17 g) was fractionated by VLC eluted with a mixture of hexane: EtOAc in increasing polarity (90:10 to 0:100), obtaining 50 fractions that were combined in 7 final fractions according to the study by TLC. Fraction 2 (42 g) was purified by successive FC eluted with DCM: EtOAc (70:30), obtaining **1** (2.2 g) and a crystalline yellow solid known as (*2′Z*) cumenic acid (**2**, 157 mg, 132–134 °C). Fractions 3 and 4 (2.9 g) were subjected to purification by successive FC eluted with CHCl_3_:MeOH (99:1), DCM:EtOAc (90:10), and Hex:CHCl_3_:MeOH 40:40:20, obtaining a yellow oil called gaudichaudianic acid (**3**, 1.2 g). Fractions 5–6 were combined (3.6 g) and subjected to purification by successive FC eluted with DCM:EtOAc (95:5 to 70:30), hexane–acetone (70:30 to 40:60), and hexane–EtOAc (90:10 to 40:60) to obtain a crystalline white solid known as 4-methoxy-3-(3′-methyl-2′-butenyl)benzoic acid (**4**, 27 mg, 127–129 °C). 

(*2′E*) *Cumenic acid*
**1**. Yellow crystals, m. p. 112–113 °C. ^1^H-NMR (CDCl_3_, 400 MHz,): δ (ppm) 13.81 (s, 1H), 8.47 (d, *J* = 2.0 Hz, 1H), 8.03 (d, *J* = 2.0 Hz, 1H), 6.85 (s, 1H), 5.34 (m, 2H), 5.14 (m, 1H), 3.39 (d, *J* = 7.3 Hz, 2H), 2.33 (d, *J* = 6.5 Hz, 2H), 2.29 (m, 2H), 2.23 (d, *J* = 1.0 Hz, 3H),1.77 (s, 3H), 1.73 (s, 6H), 1.65 (s, 3H). APT (CDCl_3_, 100 MHz): δ (ppm) 196.2 (C = O), 171.9 (C = O), 166.0 (C-4), 163.0 (C-3′), 136.0 (C-6), 134.0 (C-3″), 133.0 (C-7′), 131.5 (C-3), 130.9 (C-2), 122.8 (C-6′), 120.9 (C-2″), 119.5 (C-5), 119.1 (C-2′), 118.8 (C-1), 41.8 (C-4′), 27.6 (C-1″), 26.2 (C-5′), 25.8 (C-4″), 25.8 (C-8′), 20.3 (C-10′), 17.8 (C-5″), 17.7 (C-9′). The spectroscopic data were consistent with those reported in the literature for (*2′E*) Cumenic acid [29].

(*2′Z*) *cumenic acid*
**2**. Yellow crystals, m. p. 132–134 °C. ^1^H-NMR (CDCl_3_, 400 MHz,): δ (ppm) 13.81 (s, 1H), 8.47 (d, *J* = 2.0 Hz, 1H), 8.03 (d, *J* = 2.0 Hz, 1H), 6.85 (s, 1H), 5.34 (m, 2H), 5.14 (m, 1H), 3.39 (d, *J* = 7.3 Hz, 2H), 2.33 (d, *J* = 6.5 Hz, 2H), 2.29 (m, 2H), 2.23 (d, *J* = 1.0 Hz, 3H),1.77 (s, 3H), 1.73 (s, 6H), 1.65 (s, 3H). APT (CDCl_3_, 100 MHz): δ (ppm) 196.2 (C = O), 171.9 (C = O), 166.0 (C-4), 163.0 (C-3′), 136.0 (C-6), 134.0 (C-3″), 133.0 (C-7′), 131.5 (C-3), 130.9 (C-2), 122.8 (C-6′), 120.9 (C-2″), 119.5 (C-5), 119.1 (C-2′), 118.8 (C-1), 41.8 (C-4′), 27.6 (C-1″), 26.2 (C-5′), 25.8 (C-4″), 25.8 (C-8′), 20.3 (C-10′), 17.8 (C-5″), 17.7 (C-9′). The spectroscopic data were consistent with those reported in the literature for (*2′Z*) Cumenic acid [35].

*Gaudichaudianic acid***3**. Yellow oil, ${\left[\alpha \right]}_{D}^{25}$: +21.0 (c 0.10, CHCl_3_), ^1^H-NMR (CDCl_3_, 400 MHz,): δ (ppm) 7.73 (s, 1H), 7.58 (d, *J* = 1,5 Hz, 1H), 6.38 (d, *J* = 10.0 Hz, 1H), 5.58 (d, *J* = 9.9 Hz, 1H), 5.28 (t, *J* = 7.2 Hz, 1H), 5.10 (t, *J* = 6.9 Hz, 1H), 3.31 (t, *J* = 11.3 Hz, 2H), 2.25 (m, 2H), 1.78 (m,2H), 1.73 (m, 6H), 1.67 (s, 3H), 1.56 (s, 3H), 1.41 (s, 3H). APT (CDCl_3_, 100 MHz): δ (ppm) 172.1 (C = O), 155.8 (C-8ª), 132.6 (C-3″), 131.8 (C-7), 131.8 (C-4′), 129.5 (C-3), 128.9 (C-8), 126.7 (C-5), 123.9 (C-3′), 122.7 (C-4), 122.0 (C-2″), 121.2 (C-6), 120.2 (C-4ª), 80.0 (C-2), 41.9 (C-1′), 28.3 (C-1″), 27.2 (C-9), 25.8 (C-6′), 25.7 (C-4″), 22.8 (C-2′), 17.6 (C-5′). The spectroscopic data were consistent with those reported in the literature for Gaudichaudianic acid [37].

*4-methoxy-3-(3′-methyl-2-butenyl)benzoic acid***4**. White crystals, m. p. 127–129 °C. ^1^H-NMR (CDCl_3_, 400 MHz,): δ (ppm) 7.98 (dd, *J* = 8.5, 2.1 Hz, 1H), 7.90 (d, *J* = 1.9 Hz, 1H), 6.88 (d, *J* = 8.6 Hz, 1H), 5.31 (t, *J* = 7.3 Hz, 1H), 3.91 (s, 3H), 3.34 (d, *J* = 7.3 Hz, 2H), 1.76 (s, 3H), 1.72 (s, 3H). APT (CDCl_3_, 100 MHz): δ (ppm) 172.5 (C-7), 162.0 (C-4), 133.4 (C-3’), 131.5 (C-2), 130.4 (C-6), 130.3 (C-3), 121.7 (C-2’), 121.4 (C-1), 109.7 (C-5), 55.7 (C-8), 28.5 (C-1’), 25.9 (C-4’), 17.9 (C- 5’). The spectroscopic data were consistent with those reported in the literature for 4-methoxy-3-(3′-methyl-2-butenyl)benzoic acid [38].

### 3.4. Preparation of Analogs and Derivatives

*1-(2-Hydroxy-phenyl)-3-methyl-but-2-en-1-one (**5**) and 4-Butyl-2, 2-dimethyl-chroman-4-ol (**6**).* 2,2-dimethylbenzopyran-4-one (176 mg, 1.0 mmol) was dissolved in THF (5 mL), then, 0.7 mL of n-BuLi (1.4 M, 1.0 mmol) at a temperature of −78 °C, a round-bottom flask was added and left to react for 2 h maintaining the temperature. Subsequently, bromobutane (2.0 mmol) was added, the temperature was maintained for 4 h, and it was allowed to rise to 4 °C for 24 h. When the reaction was finished, 10 mL of 10% NH_4_Cl was added and it was extracted with EtOAc (3 × 25 mL), 10 mL of saturated NaCl was added and reextracted with EtOAc, dried with anhydrous Na_2_SO_4_, and the solvent was removed under reduced pressure. The residue obtained was purified by flash chromatography using hexane mobile phase, obtaining two products, orange oils **5** (40.7%) and **6** (57.1% yield) [53]. 

Compound **5**: Orange oil, 40.7% yield. IR (film) nmax = 3048, 2993, 1690, 1641, 1278, 840 cm^−1^. ^1^H-NMR (CDCl_3_, 400 MHz,): δ (ppm) 12.83 (s, 1H), 7.80 (dd, *J* = 8.0, 1.7 Hz, 1H), 7.46 (ddd, *J* = 8.6, 7.3, 1.6 Hz, 1H), 7.00 (dd, *J* = 8.4, 1.0 Hz, 1H), 6.89 (ddd, *J* = 8.2, 7.2, 1.2 Hz, 1H), 6.84–6.78 (m, 1H), 2.24 (d, *J* = 1.2 Hz, 3H), 2.07 (d, *J* = 1.2 Hz, 3H). APT (CDCl_3_, 100 MHz): δ (ppm) 196.3 (C = O), 163.2 (C-1), 157.9 (C-3′), 135.8 (C-2′), 129.8 (C-3), 120.6 (C-4), 120.1 (C-5), 118.6 (C-2), 118.4 (C-6), 28.2 (C-4′), 21.4 (C-5′). HRMS (ESI) calc. for C_11_H_11_O_2_ [M-H]^−^: 175.0837, found: 175.0845. 

Compound **6**: Orange oil, 57.1% yield. IR (film) nmax = 3350, 2960, 1500, 1480, 1325, 1000 cm^−1^. ^1^H-NMR (CDCl_3_, 400 MHz,): δ (ppm) 7.44 (dd, *J* = 7.8, 1.6 Hz, 1H), 7.20 (ddd, *J* = 8.2, 7.2, 1.7 Hz, 1H), 7.02–6.93 (m, 1H), 6.85 (dd, J = 8.2, 1.2 Hz, 1H), 2.10 (s, 1H), 2.01 (s, 1H), 1.96 (s, 1H), 1.82 (s, 1H), 1.46 (s, 3H), 1.40 (s, 3H), 1.34 (d, *J* = 0.7 Hz, 4H), 0.90 (t, *J* = 7.2 Hz, 3H). APT (CDCl_3_, 100 MHz): δ (ppm) 153.1 (C-8a), 129.0 (C-7), 127.5 (C-6), 126.2 (C-5), 120.8 (C-8), 118.1 (C-4a), 74.1 (C-2), 69.2 (C-4), 45.7 (C-3), 43.7 (C-1′), 30.3 (C-2′), 26.5 (C-9), 25.2 (C-10), 23.1 (C-3′), 14.1 (C-4′). HRMS (ESI) calc. for C_15_H_21_O_2_ [M-H]^−^: 233.1620, found: 233.0598.

*1-(2,2-Dimethyl-2H-chromen-6-yl)-3,7-dimethyl-octa-2,6-dien-1-ol.* (**7**). The compound 6-Bromo-2,2-dimethyl-2H-chromene (238 mg, 1.0 mM) was dissolved in dry THF (6 mL), then 0.7 mL (1 mL) at a temperature of −78 °C was added, (1.0 mmol) of 1.4 M sec-BuLi, and immediately, the geranialdehyde 151 mg (1.0 mmol) was added. The reaction mixture was left under constant stirring for 3 h at −78 °C, then 5 mL of saturated NH_4_Cl was added, it was left stirring and at room temperature for 24 h. To complete conversion, it was extracted with EtOAc (3 × 25 mL), saturated NaCl was added and reextracted with EtOAc, dried with anhydrous Na_2_SO_4_ and the solvent was removed under reduced pressure, to give **7** with 41% yield [54]. 

Compound **7**: Yellow oil, yield 41%. IR (film) nmax = 3350, 2960, 1471, 1435, 1315, 1325, 1070, 980 cm^−1^. ^1^H-NMR (CDCl_3_, 400 MHz,): δ (ppm) 7.13 (td, *J* = 7.8, 1.7 Hz, 1H), 7.00 (dd, *J* = 7.4, 1.6 Hz, 1H), 6.87 (td, *J* = 7.4, 1.0 Hz, 1H), 6.81 (d, *J* = 8.1 Hz, 1H), 6.35 (d, *J* = 9.8 Hz, 1H), 5.63 (d, *J* = 9.8 Hz, 1H), 5.29 (t, *J* = 6,9 Hz, 1H), 5.24 (t, *J* = 6,8 Hz, 1H), 4,86 (dd, *J* = 8.8, 6.2 Hz, 1H), 2.25 (m, 2H), 2.17 (t, *J* = 7.4, 2H), 1.69 (s, 3H), 1.53 (s,6H), 1.47 (s, 6H). APT (CDCl_3_, 100 MHz): δ (ppm) 152.9 (C-8a), 138.4 (C-3′), 132.2 (C-3), 131.7 (C-7), 130.7 (C-7′), 129.0 (C-5), 126.3 (C-2′), 124.2 (C-6′), 122.3 (C-6), 121.3 (C-4), 120.7 (C-4a), 116.3 (C-8), 72.0 (C-2, C-1′), 38.3 (C-4′), 28.0 (C-9, C-10), 26.5 (C-10′), 21.8 (C-8′), 18.1 (C-9′). HRMS (ESI) calc. for C_21_H_17_O_2_ [M-H]^−^: 311.2089 found: 311.2112.

*3-(3,7-Dimethyl-octa-2,6-dienoyl)-4-hydroxy-5-(3-methyl-but-2-enyl)-benzoic acid methyl ester* (**8**). The compound 1 (0.56 mmol) was dissolved in DFM (15 mL), then (4.5 mmol) of sodium bicarbonate was added, it was left stirring for 30 min, then methyl iodide was added (9 mmol). The mixture was allowed to stir and at room temperature for 24 h. It was diluted with EtOAc t and neutralized with 10% HCl and extracted with EtOAc (3 × 25 mL), saturated NaCl was added and reextracted with EtOAc, dried with anhydrous Na_2_SO_4_, and the solvent was removed under reduced pressure. The crude obtained product was purified by FC eluted with hexane:EtOAc (9:1), to give a 72% yield [55]. 

Compound **8**: Yellow oil, yield 72%. IR (film) nmax = 3394, 2970, 2924, 1689, 1635, 1604, 1500, 1442, 1280, 1026, 910 cm^−1^. ^1^H-NMR (CDCl_3_, 400 MHz,): δ (ppm) 13.69 (s, 1H), 8.37 (d, *J* = 2.1 Hz, 1H), 7.96 (d, *J* = 2.0 Hz, 1H), 6.84 (s, 1H), 5.37–5.27 (m, 1H), 5.13 (t, *J* = 6.6, 4.8, Hz, 1H), 3.90 (s, 3H), 3.37 (d, *J* = 7.3 Hz, 2H), 2.32 (s, 4H), 2.22 (d, *J* = 1.1 Hz, 3H), 1.76 (s, 3H), 1.72 (d, *J* = 3.0 Hz, 6H), 1.63 (d, *J* = 4.0 Hz, 3H). APT (CDCl_3_, 100 MHz): δ (ppm) 196.2 (C = O), 171.9 (C = O), 166.0 (C-4), 163.0 (C-3’), 136.0 (C-6), 134.0 (C-3″), 133.0 (C-7’), 131.5 (C-3), 130.9 (C-2), 122.8 (C-6’), 120.9 (C-2″),119.5 (C-5), 119.1 (2’), 118.8 (C-1), 41.8 (C-4’), 27.6 (C-1″), 26.2 (C-5’), 25.8 (C-8’), 25.8 (C-4″), 20.3 (C-10’), 17.8 (C- 5″) 17.7 (C-9’). HRMS (ESI) calc. for C_23_H_29_O_4_ [M-H]^−^: 369.2144 found: 369.2298.

### 3.5. Biological Activity–Determination of Enzymatic Inhibition against Pancreatic Lipase (PL) and α-Glucosidase (AG) 

#### 3.5.1. PL Inhibition Assay

Determination of enzymatic inhibition of the identified compounds against the PL enzyme was carried out following the methodology described in the literature with some modifications [56,57,58]. The concentrations were evaluated for compounds from 300 to 6.25 µM. Orlistat was used as a positive control. In 96-well boxes, 30 μL of extract or compound stock solution, 30 μL of PL solution (200 U/mL), and 160 μL of Tris-HCl buffer (0.1 M, pH: 8.4) were mixed. The mixture obtained was pre-incubated for 30 min at 37 °C. Subsequently, 30 μL of p-nitrophenyl dodecanoate (100 μM) were added to complete a final volume of 250 μL and incubated for 40 min at 37 °C. After the incubation time, the absorbance was measured at 405 nm. All assays were performed in triplicate in two independent assays. IC_50_ values were estimated by nonlinear regression analysis. Statistically significant differences in biological effects of inhibitors (compounds) were analyzed and compared using the ANOVA test, supplemented by Tukey HSD post hoc analysis. All reported data corresponded to the average of three repetitions ± SD, and the statistical significance considered was *p* < 0.05.

#### 3.5.2. AG Inhibition Assay

Determination of enzyme inhibition of the compounds identified against the AG enzyme was carried out following the methodology described in the literature with some modifications [58,59,60]. The concentrations were evaluated for compounds from 300 to 6.25 µM. Acarbose was used as a positive control. In 96-well boxes, 10 μL of stock solution of each extract or compound, 20 μL of GA solution (0.5 U/mL), and 200 μL of phosphate buffer (0.1 M, pH: 7.2) were mixed. The mixture obtained was pre-incubated for 30 min at 37 °C. Subsequently, 20 μL p-nitrophenyl-α-D-glucopyranoside (400 μM) was added to complete a final volume of 250 μL and incubated for 30 min at 37 °C. After the reaction time, the absorbance at 405 nm was measured. All assays were performed in triplicate in two independent experiments. IC_50_ values were estimated by nonlinear regression analysis. Statistically significant differences in the biological effects of inhibitors (compounds) were analyzed and compared using ANOVA, supplemented by Tukey HSD post hoc analysis. All reported data corresponded to the average of three repetitions ± SD, and the statistical significance considered was *p* < 0.05.

#### 3.5.3. Kinetic Study

Kinetic studies to determine the type of inhibition were performed with PL and AG using a methodology the same as that described in the inhibitory activity assays. The compounds were evaluated at 3 different concentrations according to their IC_50_ using the following values: 0.5, 1.0, and 2.0 × IC_50_. Five substrate concentrations were used for each of the enzymes in the range between 12.5 to 200 μM p-nitrophenyl dodecanoate for PL and 25 to 800 μM for p-nitrophenyl-α-D-glucopyranoside in AG. Kinetic experiments were performed in triplicate in two independent experiments. Inhibition constants (Ki) were calculated from (substrate) vs reaction rate curves using nonlinear regression of the enzyme inhibition kinetic function. Additionally, the inhibition mechanism was graphically determined by applying the double reciprocal Lineweaver–Burk regression function [56,61,62].

### 3.6. Molecular Docking Studies

Interaction, docking, and binding analyses in 3D were performed using AutoDockTools (ADT), AutoDock Vina 1.1.2 (ADV) (Maestro Release-2016 from the Schrödinger platform) [63,64]. The crystal structures of PL and AG were obtained from the Protein Data Bank website (www.rcsb.org, accessed on 1 July 2022, PDB ID: 1LPB and 2QMJ, respectively for each enzyme). The chemical structures (2D) of the ligands were processed by MarvinSketch software, obtaining the SMILES (Simplified Molecular Input Line Entry System) structures in its database. Finally, and based on the experimental data of the type of inhibition, the most probable binding sites of active compounds were evaluated, and the binding modes were analyzed and compared with the results obtained by each enzyme target. The Maestro academic software was used to generate 2D and 3D figures of the binding modes [63].

## 4. Conclusions

The present study constitutes the first report of the inhibitory effects on PL and AG of substances derived from *P. cumanense*, demonstrating the potential of prenylated benzoic acid derivatives as polypharmacological compounds of interest for the development of treatments for obesity and type 2 diabetes. The mechanism of enzymatic inhibition of isolated and synthesized compounds on the two digestive enzymes of interest was determined, allowing us to establish that **6** and **7** showed high potency and affinity as inhibitors of the catalytic activity of both PL and AG. Docking studies allowed us to predict not only the binding modes in the catalytic site of PL but also the alternative or allosteric sites of PL and AG proteins, providing insights into the competitive and allosteric regulations. The correlation between experimental kinetics and in silico studies provides valuable information to predict the main protein–ligand interactions to better understand the mechanisms of inhibition.

## Data Availability

Not applicable.

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
