# Peer review of "In Vitro and In Silico Study of the α-Glucosidase and Lipase Inhibitory Activities of Chemical Constituents from *Piper cumanense* (Piperaceae) and Synthetic Analogs"

_plants, 2022, doi:10.3390/plants11172188_

Round 1

Reviewer 1 Report

The manuscript entitled In Vitro and In Silico Study of the α-Glucosidase and
Lipase Inhibitory Activities of Chemical Constituents from Piper cumanense (Piperaceae) and Synthetic Analogs’ presents bioactive compounds, mainly prenylated benzoic acid and derivatives, and their effect on pancreatic lipase and
?-glucosidase.

The manuscript describes and discusses the isolation and identification of the secondary metabolites focusing on four prenylated benzoic acid and finally shows an enzymatic study by using some informatic tools.

The manuscript is interesting however some problems are detected in the methodology and results. In addition, there are some works related to Piper cumanense. However, they do not study this metabolic disorder approach. Some aspects to take into account are: -The collection of the sample. The material used was collected in the same season? Was there replication of the collection of the material? -The extraction was made by maceration with ethanol, right? How many days were the samples in contact with the solvent? , are the authors aware of the drawbacks of the maceration? -Why did the authors use concentration so high (1000 ?M) for AG and PL inhibition assay? -Other aspects to take into account would be related to the replications. Did authors consider enough 3 replications?. Some parts of the discussion related to kinetic needs be improved. In general, the manuscript studies a very interesting line, without great limitations, but some comments should be considered.

Author Response

Dear Reviewer

Thank you very much for the comments given. Regarding the aspects that you indicate that should be reviewed, we give an answer below:

-The plant material was collected in the same place and at the same time of year as the previous study you mention.

- In the extraction methodology described in the article, the mass/volume ratio used, the extraction time and the times to change the solvent were included. See text highlighted in yellow on page 9, lines 292 and 293. We know less than that maceration is a discontinuous method of extraction that means that it is necessary to replace the solvent at least every 48 hours and shake frequently.

- The highest concentration used for compounds was 300 uM. We missed that mistake in the manuscript. We already adjusted it in the assay methodology with the two enzymes. A concentration of 1000 ppm is used for extracts and fractions.

- In the methodology of the tests on the enzymes, specifically in lines 412 and 427, it is mentioned that two independent tests were carried out, each one with three replicates, having in each case a total of six replicates. We consider that with two independent trials that have three replicates it is appropriate to validate the results.

Reviewer 2 Report

In the manuscript presented to me for review, the Authors undertook to verify the ability to inhibit two key enzymes in the pathogenesis of chronic non-communicable diseases, which include type 2 diabetes and obesity. The research was carried out on compounds isolated from Piper cumanense and on their synthetic counterparts.

The research topic is very interesting. The work was planned in an exemplary manner, the research is detailed, the results were statistically analyzed and carefully presented in the manuscript.

I only have two small comments:

1 / in the introduction, please emphasize the novelty of the work and put forward a research hypothesis - thus, please refer to this hypothesis in the conclusions

2 / numerical values ​​- please standardize the notation or use accuracy to tenths or hundredths; please remember that the standard error must be reported with the same accuracy as the value involved

After revising, I recommend this work for publication in Plants.

Author Response

Dear Reviewer

Thank you very much for the comments given. Regarding the aspects that you indicate that should be reviewed, we give an answer below:

- In the introduction it was mentioned that until now the inhibitory effect of P. cumanense and its chemical constituents on the digestive enzymes of interest has not been reported. The conclusion was adjusted to respond to the objective of the investigation.

- The tenths and hundredths of the reported results were adjusted taking into account the indication given by you.